

# Seasonal regulation of herbivory and nutrient effects on macroalgal recruitment and succession in a Florida coral reef

Alain Duran[1], Ligia Collado-Vides[1,2] and Deron E. Burkepile[1,3]

[1] Biology Department, Florida International University, Miami, FL, United States
[2] Southeast Environmental Research Center, Florida International University, Miami, FL, United States
[3] Department of Ecology, Evolution, & Marine Biology, University of California, Santa Barbara, CA, United States

## ABSTRACT

Herbivory and nutrient enrichment are drivers of benthic dynamics of coral reef macroalgae; however, their impact may vary seasonally. In this study we evaluated the effects of herbivore pressure, nutrient availability and potential propagule supply on seasonal recruitment and succession of macroalgal communities on a Florida coral reef. Recruitment tiles, replaced every three months, and succession tiles, kept in the field for nine months, were established in an ongoing factorial nutrient enrichment-herbivore exclusion experiment. The ongoing experiment had already created very different algal communities across the different herbivory and nutrient treatments. We tracked algal recruitment, species richness, and species abundance through time. Our results show seasonal variation in the effect of herbivory and nutrient availability on recruitment of coral reef macroalgae. In the spring, when there was higher macroalgal species richness and abundance of recruits, herbivory appeared to have more control on macroalgal community structure than did nutrients. In contrast, there was no effect of either herbivory or nutrient enrichment on macroalgal communities on recruitment tiles in cooler seasons. The abundance of recruits on tiles was positively correlated with the abundance of algal in the ongoing, established experiment, suggesting that propagule abundance is likely a strong influence on algal recruitment and early succession. Results of the present study suggest that abundant herbivorous fishes control recruitment and succession of macroalgae, particularly in the warm season when macroalgal growth is higher. However, herbivory appears less impactful on algal recruitment and community dynamics in cooler seasons. Ultimately, our data suggest that the timing of coral mortality (e.g., summer vs. winter mortality) and freeing of benthic space may strongly influence the dynamics of algae that colonize open space.

Subjects Ecology, Marine Biology
Keywords Coral reef, Algae, Herbivorous fish, Nutrients, Season, Recruitment, Succession

Corresponding author
Alain Duran, adura023@fiu.edu

## INTRODUCTION

"Grazing ecosystems" are characterized by an herbivore-based food web where over 50% of primary production is processed by aquatic or terrestrial grazers (*McNauthon, 1985*; *Douglas, McNaughton & Tracy, 1998*; *Burkepile, 2013*). Coral reefs are grazing ecosystems
where herbivory and nutrient availability are often considered the primary ecological drivers of macroalgal community dynamics (*Littler & Littler, 1984*; *McCook, 1999*; *Burkepile & Hay, 2006*). Field experiments excluding herbivores have resulted in a substantial increase of both biomass and density of some macroalgal species, generating changes in species composition at the community level (*Lewis, 1986*; *Burkepile & Hay, 2009*; *Ferrari et al., 2012*).

Nutrient enrichment tends to have variable effects on macroalgal communities, possibly linked to species-specific responses of macroalgae to nutrient availability (*Larned, 1998*; *Fong et al., 2001*; *Dailer, Smith & Smith, 2012*). In particular, growth rates of some small, fast-growing species quickly peak in nutrient enriched environments (*Lapointe, Littler & Littler, 1997*; *McClanahan et al., 2004*; *Smith, Runcie & Smith, 2005*). In contrast, larger slow-growing species, typical of late stages of macroalgal community succession (e.g., *Sargassum* spp., *Amphiroa* spp.), often show weak or mixed effects in nutrient enriched areas (*McClanahan et al., 2004*; *Burkepile & Hay, 2009*). However, abundance of different macroalgal species can vary seasonally on reefs (*Lirman & Biber, 2000*; *Collado-Vides, Rutten & Fourqurean, 2005*; *Collado-Vides et al., 2007*), which could explain the variable relative effect of herbivores and nutrients on structuring macroalgal communities (*Burkepile & Hay, 2006*; *Smith, Hunter & Smith, 2010*).

A number of factors may influence seasonality of algal communities on reefs including seasonal changes in abiotic conditions (e.g., temperature and light; *Clifton & Clifton, 1999*), the timing and intensity of disturbances (*Diaz-Pulido & Garzon-Ferreira, 2002*; *Diaz-Pulido & McCook, 2004*; *Goodsell & Connell, 2005*) and propagule supply and recruitment (*McClanahan, 1997*; *Diaz-Pulido & McCook, 2002*; *Walters et al., 2002*; *Vroom et al., 2003*). Increases in water temperature and light availability can promote macroalgal growth and trigger reproduction in some species (*Clifton, 2008*; *Collado-Vides et al., 2011*). For example, both *Dictyota pulchella* and *Sargassum* spp. show a peak of abundance during the summer and a loss of biomass in the coolest seasons (*Lirman & Biber, 2000*; *Renken et al., 2010*). In contrast, *Gracilaria* spp. and *Stypopodium zonale* rather exhibit higher abundance during cooler seasons (*Hay & Norris, 1984*; *Chung et al., 2007*). In addition to influencing macroalgal growth rates, temperature also influences the rate of herbivory in fishes with grazing rates often peaking during warmer periods (*Ferreira, Peret & Coutinho, 1998*; *Smith, 2008*; *Lefevre & Bellwood, 2010*). Thus, variation in abiotic controls of both algal growth rates and rates of herbivory across seasons could result in temporal fluctuations of bottom-up and top-down forcing.

These seasonal differences in macroalgal growth and herbivory rates could affect how disturbances to reefs impact macroalgal community development and succession. For example, in reefs in the Florida Keys, both extreme warm water (*Eakin et al., 2010*) and cold water (*Lirman et al., 2011*) anomalies can lead to coral mortality. Given that these disturbances open up free space for macroalgal colonization during different times of the year with different abiotic conditions, different species of algae may become dominant and drive different successional trajectories depending on the timing of these disturbances and the initiation of algal succession.

Propagule abundance can also impact community dynamics by influencing the rates of recruitment in marine organisms (*Stiger & Payri, 1999*; *Lotze, Worm & Sommer, 2000*;

*Walters et al., 2002*; *Grorud-Colvert & Sponaugle, 2009*). The abundance of adult macroalgal individuals, the number of propagules they produce, and the distance to a suitable substrate for colonization can determine the number of macroalgal recruits in a given area (*Kendrick & Walker, 1991*; *Stiger & Payri, 1999*; *Lotze, Worm & Sommer, 2000*; *Yñiguez, Mcmanus & Collado-Vides, 2015*). Thus, abundant adult macroalgae might increase local macroalgal recruitment, especially after relatively localized disturbances such as coral mortality events (*Walters et al., 2002*; *Roff & Mumby, 2012*). Consequently, increases in macroalgae due to reductions in herbivory or increases in nutrient availability could lead to increased macroalgal propagule supply and a positive feedback on macroalgal abundance. Yet, no studies have directly addressed how seasonality and propagule supply interact with herbivory and nutrient availability to impact macroalgal community development and succession on coral reefs.

In the current study, we tested how both nutrient availability and herbivory varied across seasons as drivers of recruitment and succession of a coral reef macroalgal community. We used coral limestone tiles as primary substrate in a factorial field experiment, previously established for two years, manipulating access by herbivorous fishes and nutrient availability to examine the effects of herbivory, nutrient enrichment, and macroalgal abundance on algal recruitment patterns and primary succession across seasons. We quantified algal abundance and diversity on both primary substrate and the established macroalgal communities regularly over nine months. We predicted that macroalgal recruitment would be higher in areas with greater adult macroalgal abundance because adult macroalgae act as propagule supply source; and because herbivores consume late successional species, usually large palatable species, herbivory would be the main driver of macroalgal communities at late successional stages. In addition, we hypothesized that the effect of both nutrient availability and herbivory would be reduced in cooler seasons because lower temperature and less light availability reduce rates of macroalgal growth and at low temperature herbivory rates are typically lower.

## MATERIALS AND METHODS
### Study site and experimental design
This study was conducted from September 2011 to June 2012 on a spur and groove reef system located in the upper Florida Keys near Pickles Reef (25°00′05″N, 80°24′55″W) with the approval of the Florida Keys National Marine Sanctuary (FKNMS-2009-047 and FKNMS-2011-090). The reef is a shallow area (5–6 m) where parrotfishes and surgeonfishes are the dominant herbivorous fishes and the long-spined urchin, *Diadema antillarum,* is present at very low densities (<1 individual per 50 m$^2$, A Duran & D Burkepile, pers. obs., 2011). Water temperature oscillates seasonally from ∼24 °C in winter (December and January) to over 30 °C during summer (Appendix S1).

In June 2009, eight 9 m$^2$ experimental plots (3 × 3 m) were established to examine the interactive effects of herbivory and nutrient availability on benthic community dynamics (*Zaneveld et al., 2016*). Plots were separated by at least 5 m. Each 9 m$^2$ plot contained two quadrats (1 × 1 m$^2$) for herbivore exclusion (exclosure), covered with plastic-coated wire mesh (2.5 cm diameter holes) around a 0.5 m high metal bar frame. Two other quadrats

$(1 \times 1 \text{ m}^2)$ were used as herbivore exclusion controls (uncaged) that had metal bar frames with only three sides covered with wire mesh to allow access to all herbivores. As part of the ongoing experiment on the benthos, a third (completely open, no sides or top) control was also used and showed no differences in algal communities between the exclusion controls and completely open areas (*Zaneveld et al., 2016*).

To mimic nutrient loading, four of the eight 9 m² experimental plots were enriched with Osmocote (19-6-12, N-P-K) slow-release garden fertilizer. The Osmocote (175 g) was placed in 5 cm diameter PVC tubes with 10 (1.5 cm) holes drilled into them. These tubes were wrapped in fine plastic mesh to keep the fertilizer inside and attached to a metal nail within the plot for a total of 25 enrichment tubes spread evenly across each enrichment plot. Enrichment tubes were replaced every 4–6 weeks to ensure continual nutrient addition. The other four 9 m² plots were kept at ambient nutrient conditions. Yearly sampling of water column nutrients of each experimental plots (Summer 2009, 2010 and 2011) showed that this enrichment increased both dissolved inorganic nitrogen (3.91 µM vs. 1.15 µM in enriched vs. ambient) and soluble reactive phosphorus (0.27 µM vs. 0.035 µM in enriched vs. ambient) in the water column (for detailed methods see *Vega-Thurber et al. (2014)*). Additionally, nitrogen concentration in the tissues of the common alga *Dictyota menstrualis* collected from each plot every year (Summer 2009, 2010 and 2011) was 20% higher in the enriched plots compared to the control plots, suggesting that the nutrients from the enrichment were consistently available to benthic organisms (*Vega-Thurber et al., 2014*). The levels of DIN and SRP in the enriched treatment were similar to those reported from other anthropogenic-impacted reefs located around the world (*Dinsdale et al., 2008*).

Ambient nutrient values were obtained from the SERC-FIU Water Quality Monitoring Network, data from this network is open to the public (http://serc.fiu.edu/wqmnetwork/FKNMS-CD/DataDL.htm). Levels of both dissolved inorganic nitrogen (DIN) and soluble reactive phosphorus (SRP) in the ambient nutrient plots were within the range of concentrations for offshore reefs in the Florida Keys (*Boyer & Briceño, 2010*). Water quality data from a nearby reef (Molasses Reef) shows a high variability of nutrient concentration along the years. Particularly, during the study period, ambient levels of DIN (mean 0.58 µM, max 0.87 µM) and SRP (mean .038 µM, max 0.070 µM) were consistently lower than the enrichment treatments (Appendix S1). The highest values of DIN, SRP and temperature were reported for the end of May 2012 and lower values were detected in April 2011 for DIN and December 2011 for SRP, while temperature had its lowest in December 2011 (Appendix S1).

## Fish community structure

To estimate the intensity of herbivore pressure, fish community structure was evaluated four times during the study period (September, 2011; January, 2012; April, 2012 and July 2012) using $30 \times 2$ m belt transects ($n = 12$) placed haphazardly across the study site following AGRRA methodology (Protocols Version 5.4; *Lang et al., 2010*). All individuals of all fish species included in the AGRRA protocol were identified and size estimated to the nearest cm by same diver. Size estimates were converted to biomass for each individual fish using published length: weight relationships (*Bohnsack & Harper, 1988*). We did not quantify abundances of the urchin *D. antillarum* as they have been rare across the Florida
Keys (*Chiappone, Swanson & Miller, 2002*; *Chiappone et al., 2008*) and they have been very infrequently seen at our field site since the establishment of the experiment in 2009 (D Burkepile, pers. obs., 2011).

## Recruitment of macroalgae on primary substrate

To study macroalgal recruitment across different seasons in the different treatments, we placed two coral limestone settlement tiles ($10 \times 10$ cm; cut from quarried South Florida Pleistocene limestone) in each of the two exclosure and uncaged quadrats within every 9 m$^2$ plot ($n = 64$ tiles total) in September 2011. We did not put tiles in completely open areas as data from the main experiment showed that the macroalgal communities in the uncaged and completely open areas did not differ (*Zaneveld et al., 2016*). The tiles were attached to plastic mesh with cable ties and anchored to the benthos using 1.9 cm galvanized staples. These tiles (hereafter 'recruitment tiles') were collected after three months and replaced with new tiles to quantify recruitment and early succession during each season. These deployments resulted in a total of three separate sets of data on macroalgal recruitment across different seasons: fall (September–December 2011), winter (December 2011–March 2012) and spring (March–June, 2012). A tropical storm in summer 2012 removed much of the experimental infrastructure precluding data from the planned summer period.

After three months in the field, recruitment tiles were transported to the laboratory where algae were identified to the lowest possible taxonomic level (Appendix S2). Their percent cover was visually quantified using a rule to measure percent cover of each algal individual. Abundance of each species was classified from 0.1 (single individual <0.5% cover), 0.5 (less than three sparse individuals and <1% cover), 1 (few individuals and <5% cover), and then 5–100 with multiples of 5 by the same trained specialist. If a single individual covered 10% of the tile, then that individual got a 10% cover. If a single individual was present but below 1% cover, it was scored as 0.1 or 0.5% cover depending upon size. The recruitment tiles were then placed in individual separate aquaria to avoid possible propagule contamination. Each tank was prepared to replicate the field conditions as closely as possible (salinity: 35–36 ppt using artificial seawater, temperature: 25–28 °C, constant water circulation, and artificial high output white light with 12:12 day-night cycle). Initial water was filtered and then the tanks were re-fill with artificial water (Instant ocean) to avoid propagule contamination from the ocean. We kept the tiles in their corresponding aquaria for three months to promote growth of macroalgal recruits that were unidentifiable in our immediate evaluation due to their small size or lack of identifiable traits. After this period, all macroalgal species were re-identified and any new contribution was added to the species list.

## Succession of macroalgal communities on primary substrate

In September 2011, we also placed a second set of two coral limestone settlement tiles ($10 \times 10$ cm) (hereafter 'succession tiles') in each exclosure and uncaged quadrat ($n = 64$ tiles total). Succession tiles were kept in the field from September 2011 to June 2012. Macroalgal abundance was visually quantified on succession tiles *in situ* in January and June 2012 using the same method described above. Macroalgae were identified to the lowest taxonomic level possible and also binned into form-functional groups (FFG) following *Steneck & Detheir (1994)*.

## Established macroalgal communities

Algal abundance of established communities showed significant differences in the benthic macroalgal community composition across the different treatments (*Zaneveld et al., 2016*). These differences in the abundance and community composition of algae could have resulted in differing levels of propagule abundance across treatments, an important factor potentially affecting recruitment and succession on primary substrate in our study. To evaluate the potential propagule supply of each established community, macroalgal abundance was visually quantified within each exclosure and uncaged quadrat in January and June 2012 using the percent cover scale and FFG classification as described above.

## Statistical analyses

Average values of biomass, density and percent cover are followed by calculated standard error. Biomass and density of total and herbivorous fish were compared across seasons using a one-factor ANOVA. For statistical analyses of the different macroalgal community metrics of recruitment and succession tiles, we averaged data from the two tiles located within each exclosure and uncaged quadrat. For recruitment tiles, succession tiles, and established communities we averaged metrics of the two exclosure quadrats and two uncaged quadrats of each plot such that $n = 4$ for each treatment except for the ambient-exclosure treatment where $n = 3$ due to losing exclosures in one plot in May 2012 during a storm.

To test for the effects of herbivores, nutrient enrichment, season, and their interactions on algal species richness and overall macroalgal abundance of recruitment tiles, we used a split plot ANOVA with herbivory treatments as subplots nested within nutrient treatment plots, when there were significant treatment X season interactions, we used split plot ANOVA to assess treatment effects (i.e., nutrient enrichment and herbivore exclosure) within different seasons. Post-hoc analysis Tukey HSD was used to test for differences across season. We used non-metric multi-dimensional scaling (nMDS) and permutational MANOVA (PERMANOVA) to assess the effects of treatments and seasonality on macroalgae community composition of recruitment tiles. We used a similarity percentage analysis (SIMPER) to assess how different species contributed to differences in community structure across treatments. To assess variability in abundance of most common species across treatments and seasons, we used split plot ANOVAs or non-parametric tests when data did not satisfy assumptions for parametric tests.

To test the effects of herbivory, nutrient availability, and season on overall algal abundance and the abundance of different FFG, for both successional tiles and established algal communities, we used a split plot ANOVA, with herbivory treatments as subplots nested within nutrient treatment plots. To test the effects of treatment on community succession, a non-metric multi-dimensional scaling (nMDS) and a PERMANOVA were performed on the abundance of all FFG analyzed seasonally. To examine how macroalgal abundance in the established communities (potential propagule supply) impacted macroalgal recruitment, we used a Pearson correlation to assess the relationship between FFG abundance of established communities and both, succession tiles and recruitment tiles, in both winter and spring. We performed descriptive and inferential analyses using packages

Vegan, doBy, MASS and ggplot2 from R program created by *R Development Core Team (2012)*, version 3.2.2.

## RESULTS

### Fish community structure

Overall fish mean biomass and density at the study site were $6495.60 \pm 508.10$ g/100 m$^2$ (mean $\pm$ SE; data presented as such hereafter) and $39.93 \pm 3.20$ Ind./100 m$^2$, respectively. Herbivores (family Scaridae and Acanthuridae) comprised 78% of overall fish biomass with an average of $5087.17 \pm 569.50$ g/100 m$^2$ and 74% of overall fish density $29.93 \pm 2.10$ Ind./100 m$^2$. Total biomass of parrotfish and surgeonfish were $2771.65 \pm 526.60$ g/100 m$^2$ and $2315.52 \pm 370.60$ g/100 m$^2$, respectively. We saw no temporal changes in biomass or density of total and herbivorous fish as no significant differences were found among seasons (One-factor ANOVA, $p > 0.05$ in all cases, Appendix S3).

### Recruitment of macroalgae on primary substrate (recruitment tiles)

We identified 101 macroalgal taxa (Appendix S2) including field and laboratory observations. Macroalgal species richness on recruitment tiles increased across seasons, averaging $9.73 \pm 0.63$ species per tile in fall, $12.13 \pm 0.79$ in winter, and $14.40 \pm 1.19$ in spring (split plot ANOVA, Season, $p = 0.003$). Neither nutrient enrichment nor herbivore exclosure had an independent or interactive effect on species richness of recruitment tiles (Appendix S2). Overall abundance of macroalgae on recruitment tiles was twofold higher in spring ($116.12 \pm 9.50$%) compared with fall ($60.00 \pm 7.48$%) and winter ($51.77 \pm 6.31$%) regardless of treatment (Fig. 1, split plot ANOVA, Season, $p = 0.02$). Across seasons the combination of herbivore exclosure and nutrient enrichment had significant impact with noticeable increase in macroalgal abundance compared to other treatments (Fig. 1; Appendix S3).

Macroalgal assemblages on recruitment tiles were different across seasons (nMDS, Fig. 2, PERMANOVA, Season: pseudo $F = 7.68$, $p = 0.01$). Only four groups were present in all seasons (crustose coralline algae (CCA), Cyanobacteria, *Jania capillacea* and *Peyssonnelia* spp.) but with dissimilar abundances (Table 1). There was a peak of cyanobacteria in spring while the abundance of *Peyssonnelia* spp. was four times higher during fall and spring compared to winter (Table 1). Other species such as *Ectocarpus* sp., *Gelidiella* sp. and *Heterosiphonia* sp. increased their abundance in winter although abundance of both *Laurencia* species peaked in spring (Table 1). There was an effect of herbivore exclosure and a significant interaction between herbivore exclosure and season in driving differences in community composition on recruitment tiles (PERMANOVA, Herbivore: pseudo $F = 3.94$, $p = 0.01$ and Herbivory:Season interaction: pseudo $F = 2.15$, $p = 0.01$, respectively). Analyses within season showed a clear effect of herbivore exclosure and nutrient enrichment in spring which seems to be stronger when both are combined as shown in the nMDS analyses (Fig. 2; PERMANOVA, Herbivory: pseudo $F = 6.16$, $p = 0.01$ and Nutrient: pseudo $F = 3.08$, $p = 0.04$, respectively).

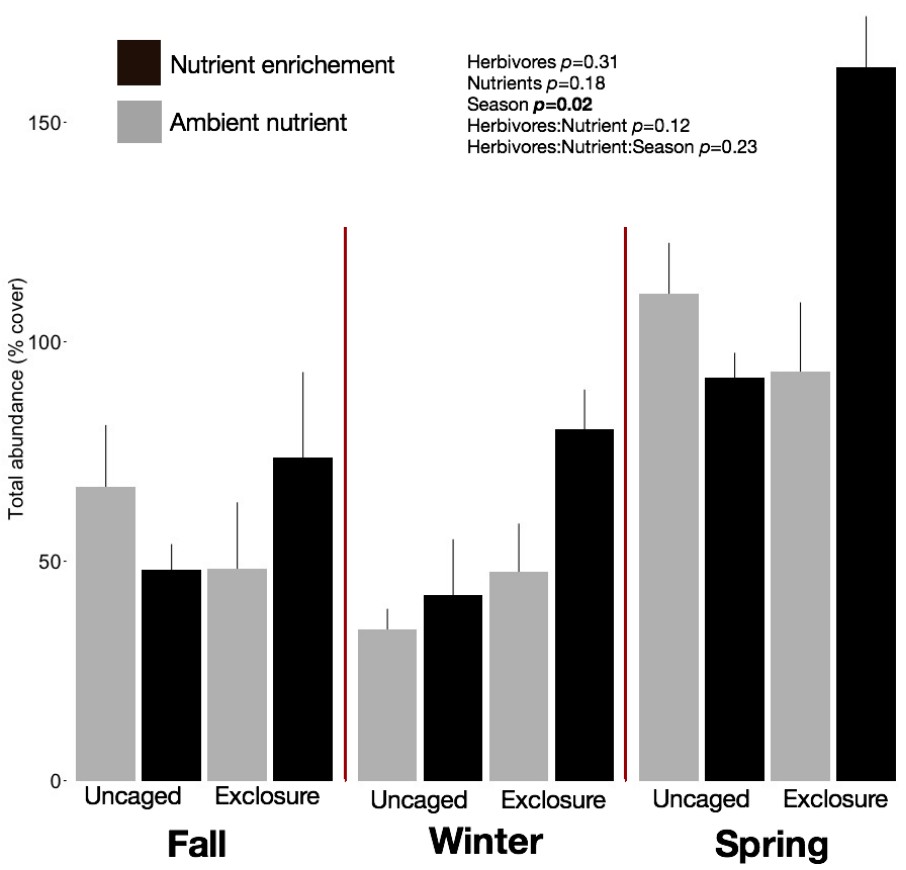

**Figure 1** **Overall abundance of macroalgae on recruitment tiles by treatments within each season.** Bars represent means ± SEM. Probability values (*p*) come from Split plot ANOVA of main treatment effects. Red lines help to distinguish bars across seasons.

## Succession of macroalgae on primary substrate (succession tiles)

Excluding herbivores from succession tiles during the study period led to almost double overall macroalgal cover (77.29 ± 7.29%) compared to uncaged tiles (40.84 ± 5.22%; split plot ANOVA, Herbivory, $p = 0.03$), while no other factors showed significant effects (Fig. 3; Appendix S2). Filamentous algae increased abundance in spring with 33.83 ± 4.58% and was negatively affected by nutrient enrichment (Fig. 3, split plot ANOVA, Nutrient, $p = 0.05$). Herbivory exclusion in ambient nutrient plots increased abundance of filamentous algae in winter while decreased it in spring (Fig. 3, split plot ANOVA, Herbivory:Nutrient:Season interaction, $p = 0.03$). Abundance of foliose macroalgae (e.g., *Dictyota* spp.) increased when herbivores where excluded (Fig. 3; split plot ANOVA, Herbivory, $p = 0.03$), there was a trend towards nutrient enrichment decreasing foliose macroalgae (Fig. 3; split plot ANOVA, Nutrient interaction, $p = 0.08$). Leathery algae (e.g., *Sargassum* spp.) increased in June (Fig. 3, split plot ANOVA, Season, $p = 0.05$) and were almost exclusively present in herbivore exclosures, although their abundance was variable and we did not detect any significant statistical effect of exclosures (Fig. 3). Articulated calcareous

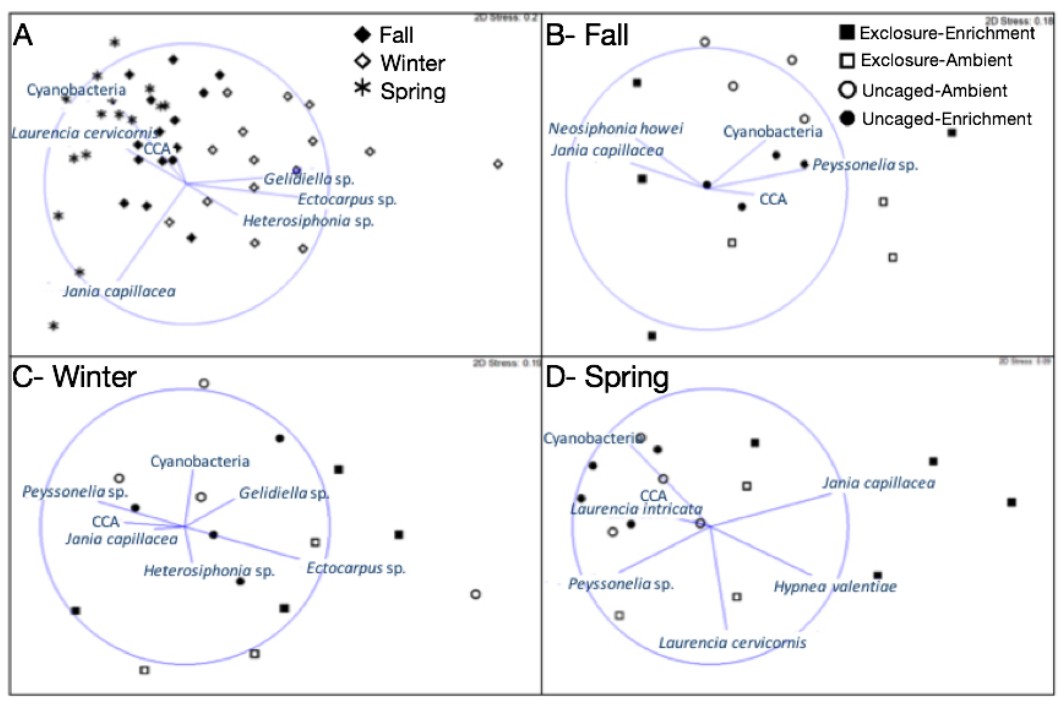

**Figure 2** **Non-metric multidimensional scaling analysis of algal communities on recruitment tiles across seasons (A) and in each treatment by season (B–D).** Analysis was performed using percent cover of algal species among treatments.

**Table 1** **Abundance of dominant macroalgal taxa found on recruitment tiles per season.** Statistics column refers to analysis performed to compare abundance of specific taxa across season ($F$ and $p$ from one-factor ANOVA and N/A and $p$ from Friedman test). Different letters indicate post hoc analysis (SNK or Wilcoxon pair analysis) when abundance differed across seasons.

| Species | Fall (Sep–Dec) | Winter(Jan–Mar) | Spring (Mar–Jun) | Statistical sign. | |
|---|---|---|---|---|---|
| | Percent cover (%) | Percent cover (%) | Percent cover (%) | F | $p$ |
| *Peyssonnelia* sp. | 12.28 (A) | 3.43 (B) | 12.36 (A) | 11.405 | **0.001** |
| Crustose coralline algae (CCA) | 11.24 | 7.41 | 6.40 | 2.157 | 0.124 |
| *Jania capillacea* | 4.68 | 2.20 | 10.44 | N/A | 0.173 |
| *Neosiphonia howei* | 3.90 (A) | 0.00 (B) | 2.86 (A) | N/A | **0.001** |
| Cyanobacteria | 6.6 (B) | 3.56 (B) | 37.84 (A) | N/A | **0.001** |
| *Heterosiphonia* sp. | 0.00 (B) | 1.27 (A) | 0.00 (B) | N/A | **0.001** |
| *Ectocarpus* sp. | 0.00 (B) | 12.20 (A) | 0.00 (B) | N/A | **0.001** |
| *Laurencia cervicornis* | 2.28 (B) | 0.33 (B) | 6.48 (A) | N/A | **0.001** |
| *Hypnea spinella* | 1.56 (A) | 0.82 (A) | 4.30 (B) | N/A | **0.001** |
| *Gelidiella* sp. | 0.00 (B) | 1.16 (A) | 0.00 (B) | N/A | **0.001** |
| *Laurencia intricata* | 0.00 (A) | 0.00 (A) | 3.42 (B) | N/A | **0.001** |

algae (e.g., *Jania* spp. and *Amphiroa* spp.) were only present in exclosure treatments (Fig. 3) showing a strong effect of herbivore exclusion (the split plot ANOVA did not pick up this signal likely due to all zeros in the uncaged areas). Nutrient enrichment also facilitated articulated calcareous algae with a trend towards higher abundance in nutrient enriched herbivore exclosures (Fig. 3, split plot ANOVA, Herbivory: Nutrient interaction, $p = 0.06$).

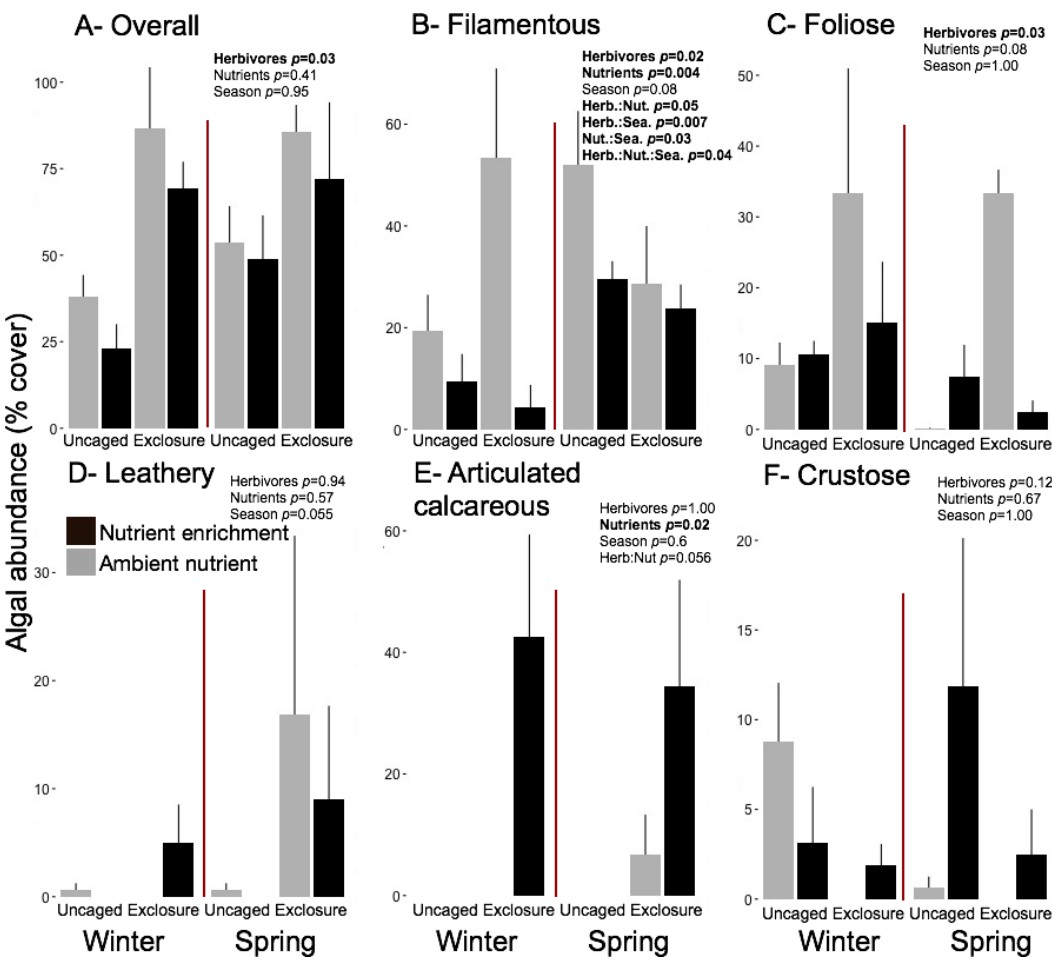

**Figure 3** **Abundance of macroalgal FFG on succession tiles by treatment in Winter (January) and Spring (June).** Bars represent means ± SEM. Probability values ($p$) come from Split plot ANOVA for main treatment effects and significant interactions. Red lines help to distinguish bars across seasons with right side for spring and left side winter.

In both winter and spring, the nMDS analysis showed herbivore exclosure had significant effects on the FFG composition of macroalgal communities (Fig. 4, PERMANOVA, Herbivory: pseudo $F = 8.96$; $p = 0.01$, pseudo $F = 3.46$, $p = 0.03$ respectively). However, there was an effect of nutrient enrichment only in January (Fig. 4, PERMANOVA, Nutrient: pseudo $F = 2.84$; $p = 0.03$).

### Established macroalgal communities

Overall macroalgal abundance of established communities was over twofold higher in spring with $84.3 \pm 7.76\%$ compared to winter $39.36 \pm 7.55\%$ (Fig. 5, split plot ANOVA, Season, $p = 0.05$). Herbivore exclosures had two fold higher algal cover (Fig. 5, split plot ANOVA, Herbivory, $p = 0.006$) than uncaged treatment while there was no effect of nutrient enrichment (Fig. 5, Appendix S3). Filamentous algae were the only macroalgal group that showed a seasonal increase from winter to spring on established communities (Fig. 5, split plot ANOVA, Season, $p = 0.04$). The three groups of upright macroalgae: foliose,

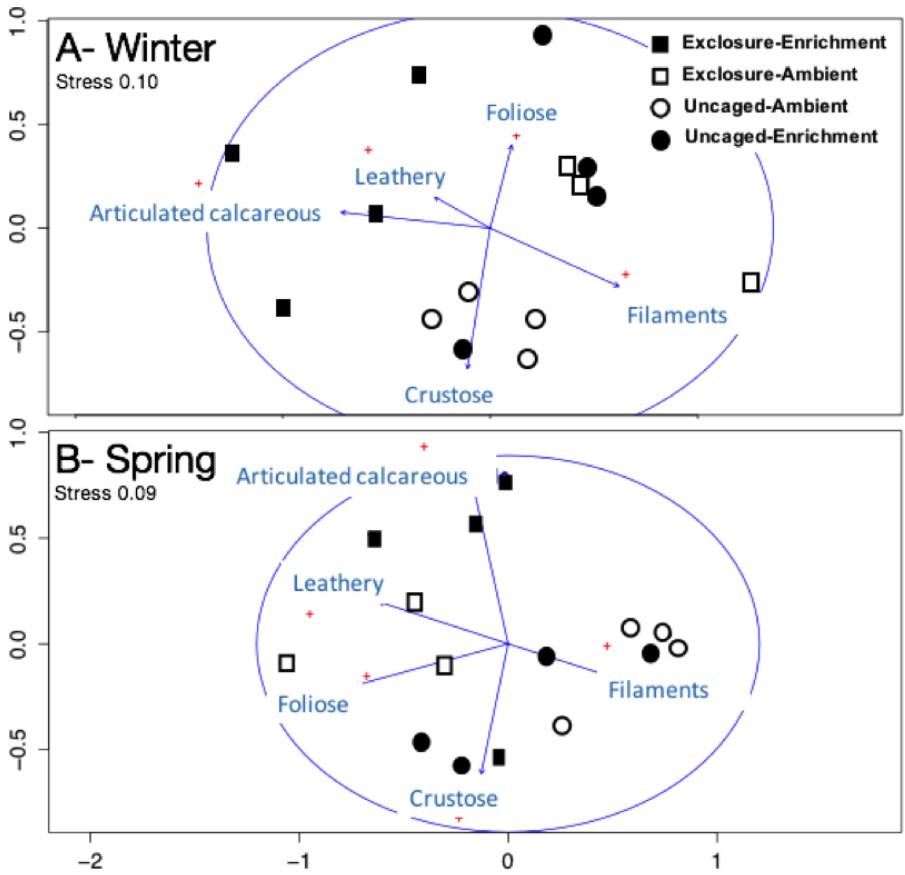

**Figure 4  Non-metric multidimensional scaling analysis of algal communities on successional tiles.**
Analysis was performed using percent cover of form-functional groups of algae among treatments in winter (A) and spring (B).

leathery and articulated calcareous algae were more abundant in herbivore exclosures (Fig. 5). Leathery macroalgae were practically only found in herbivory exclosure treatments regardless of nutrient treatment (Fig. 5). Furthermore, articulated calcareous algae (e.g., *Jania* spp. and *Amphiroa* spp.) were much more abundant inside exclosures when combined with nutrient enrichment in both winter and spring compared to ambient nutrient levels (Fig. 5, split plot ANOVA, Herbivory:Nutrient interaction, $p = 0.056$).

There were significant positive correlations of algal abundance of established communities with algal abundance found on recruitment and succession tiles for some algal groups (Table 2). The abundance of leathery macroalgae on established communities was correlated with the corresponding abundances of each treatment found on recruitment tiles in winter (Pearson correlation, $r = 0.97$, $p = 0.03$) and with abundance of succession tiles in spring (Pearson correlation, $r = 0.95$, $p = 0.05$). Articulated calcareous algae was the only algal group that showed correlations between established communities and corresponding recruitment and succession tiles in both seasons (Table 2).

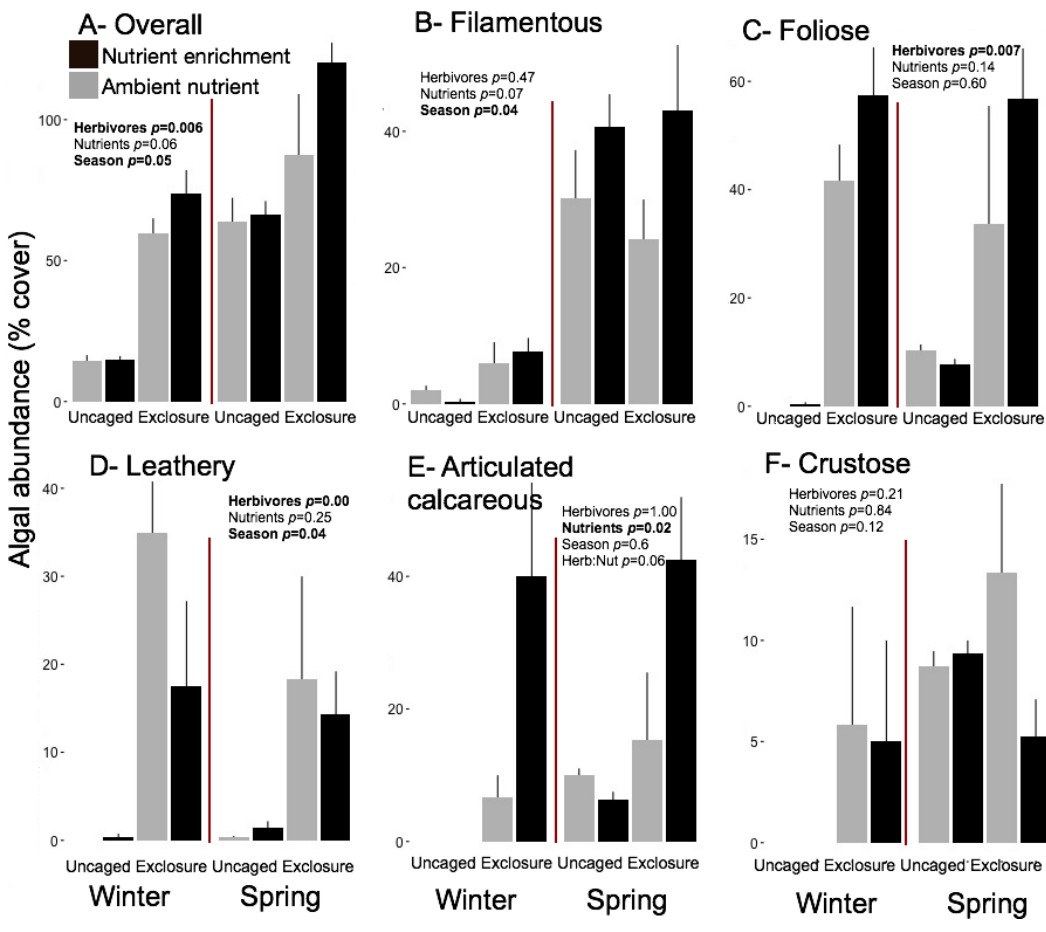

**Figure 5 Abundance of macroalgal form-functional group on established communities by treatment in winter and spring.** Bars represent means ± SEM. Probability values (*p*) come from Split plot ANOVA of main treatment effects and significant interactions. Red lines help to distinguish bars across seasons with right side for spring and left side for winter.

## DISCUSSION

Here, we show that the impact of herbivory and nutrient availability on recruitment of coral reef algae changes across seasons. We observed an increase in species richness and abundance of macroalgal recruits towards the warmer (Appendix S1) season (spring), with recruit abundance noticeably higher when combining reduced herbivory and nutrient enrichment. However, herbivory primarily drove macroalgal abundance and the trajectory of succession over longer time periods, with higher algal abundance for some groups of macroalgae (e.g., articulated calcareous algae) under elevated nutrient conditions. We also found positive correlations between algal abundance in established communities and algal abundance on both recruitment and successional tiles. These data suggest an important role of propagule supply in influencing algal recruitment and succession. Ultimately, the impact of herbivores and nutrient availability on macroalgal recruitment and succession varied across functional groups of algae and seasons, with herbivory the strongest force in warmer seasons while nutrient availability showed the strongest effects in cooler seasons.
**Table 2  Pearson correlation between algal form functional group (FFG) abundance on recruitment or succession tiles and abundance of algal FFG in established communities for winter and spring.** Bolded values show significant effects.

| Season | FFG | Recruitment tiles | | Succession tiles | |
|---|---|---|---|---|---|
| | | Coef. corr (r) | *p* | Coef. corr (r) | *p* |
| **Winter (January)** | Overall | **0.59** | **0.002** | **0.86** | **0.001** |
| | Filamentous | 0.89 | 0.11 | 0.35 | 0.65 |
| | Foliose | 0.49 | 0.51 | 0.44 | 0.56 |
| | Leathery | **0.97** | **0.03** | 0.28 | 0.72 |
| | Articulated-calcareous | **1.00** | **0.004** | **0.99** | **0.01** |
| | Crustose | **−0.94** | **0.04** | −0.77 | 0.23 |
| **Spring (June)** | Overall | **0.53** | **0.01** | **0.43** | **0.03** |
| | Filamentous | −0.33 | 0.67 | −0.51 | 0.49 |
| | Foliose | −0.09 | 0.91 | N/S | N/S |
| | Leathery | 0.88 | 0.12 | **0.95** | **0.05** |
| | Articulated-calcareous | **0.95** | **0.05** | **1.00** | **0.01** |
| | Crustose | 0.47 | 0.53 | 0.12 | 0.88 |

Macroalgal communities on coral reefs can show noticeable temporal changes in species composition and abundance, associated with abiotic (e.g., temperature, light) and biotic factors (*Tsai et al., 2005*; *Renken et al., 2010*; *Page-Albins et al., 2012*). Some common species in the Caribbean such as *Dictyota* spp. and *Halimeda* spp. increase in abundance during summer, often covering over 50% of the benthos, while others (e.g., turf-forming species) are more abundant during cooler periods of the year (*Lirman & Biber, 2000*; *Renken et al., 2010*; *Ferrari et al., 2012*). In our study, we found an increase in overall abundance of macroalgal recruits towards spring with distinct species flourishing within treatments. For instance, recruitment of *Jania capillacea* and *Hypnea spinella* was higher in spring but mostly within exclosures, which suggests the strong control of herbivory of both species. Both *Laurencia cervicornis* and *L. intricata* increased in abundance in spring. However, *L. cervicornis* was abundant in uncaged treatments while *L. intricata* was abundant in exclosure treatments. Some species of *Laurencia* are chemically defended against herbivores (*Nagle & Paul, 1998*; *Pereira, Cavalcanti & Texeira, 2000*), which could explain the proliferation of some *Laurencia* spp. in the presence of herbivores. In contrast, the abundance of small filamentous algal species commonly consumed by herbivorous fish (e.g., *Ectocarpus* sp., *Gelidiella* sp. and *Heterosiphonia* sp.) increased in winter when other studies have shown that grazing rates often decline (*Ferreira, Peret & Coutinho, 1998*; *Lefevre & Bellwood, 2010*). Indeed, we saw an interaction between herbivory and season on community composition suggesting that herbivores have stronger effects on algal recruitment during warmer periods (spring and summer). Since recruitment of corals is often higher during spring and summer (*Van Woesik, Lacharmoise & Koksal, 2006*), the strong top-down control of algal recruitment during this period could indirectly enhance coral recruitment by freeing space for corals. Herbivory may have been less important in colder seasons due to lower grazing rates combined with lower recruitment rates and slower growth rates of algae.

Succession in the absence of herbivores is expected to follow a trajectory characterized by replacement of early, fast growing species (e.g., *Enteromorpha* sp. *Ceramium* sp., *Felmania* sp.) by late successional species such as leathery and calcareous articulated species (*McClanahan, 1997*). Our results show that nutrient enrichment and herbivore exclosure interact to drive macroalgal succession at early stages (four months), while herbivory appears more important at later stages. After four months, filamentous and foliose algae increased inside exclosures with ambient nutrient levels, while leathery and articulated calcareous flourished in exclosures with nutrient enrichment. After nine months, species considered later successional species (e.g., *Sargassum* sp. and *Amphiroa* sp.) were present almost exclusively on succession tiles in exclosures regardless of nutrient enrichment. These results suggest that nutrient availability facilitates the rapid colonization and growth of leathery and articulated calcareous algae. But, over the long term, herbivory is the primary driver of their abundance. Other studies have shown that nutrient loading does not affect macroalgal species composition at late successional stages but facilitates abundance of early successional species such as turf forming algae and cyanobacteria (*McClanahan, Carreiro-Silva & Dilorenzo, 2007*). In our study we found that both nutrient availability and herbivory are significant drivers at early successional stages, whereas nutrient showed significant effect over later successional stages only when herbivores were excluded.

Competition among algae may also be important for determining successional trajectories, especially when herbivory is low. Macroalgal communities on succession tiles within herbivore exclosures were dominated by calcareous articulated and leathery species by the end of the experiment. These species appeared to replace *Dictyota* spp. and other foliose and filamentous algae, especially under nutrient enrichment, suggesting that these late successional species are better competitors in absence of herbivores. Thus, selective grazing by herbivores on more palatable species (e.g., articulated calcareous) might facilitate the colonization and establishment of less palatable foliose algae. Coral reef herbivores often consume macroalgal species of late successional stages such as leathery (e.g., *Sargassum* spp. and *Turbinaria* spp.) and calcareous articulated (e.g., *Amphiroa* spp., *Halimeda* spp. and *Jania* spp.) (*Lobel & Ogden, 1981*; *Burkepile & Hay, 2008*; *Hoey & Bellwood, 2011*), keeping macroalgal communities in stages of early succession. *Hixon & Brostoff (1996)* found similar results where removal of grazers led to a rapid shift from green and brown filamentous algae to finely branched filaments followed by species forming thicker filaments (e.g., *Tolypoicladia glomerulata*). Similarly, *Thacker, Ginsburg & Paul (2001)* reported a community shift from unpalatable to palatable species of algae when herbivores were excluded from coral reefs on Guam. This pattern is also common in terrestrial ecosystems where selective herbivores target palatable, but often competitively superior plant species, and release unpalatable species from competition (*Briske & Hendrickson, 1998*; *Torrano & Valderrabano, 2004*).

We found that macroalgal abundance on recruitment and succession tiles was correlated with abundance of algae in established communities, which suggests that local propagule supply from the established community may impact early community development. However, we could not conclusively say that recruits were from local sources (i.e., from algae within exclosures) as opposed to from sources from greater distance. In temperate marine communities, particularly for fast-growing species of macroalgae (e.g., *Cladophora*

sp., *Polysiphonia* sp. and *Ceramium* sp.), propagule abundance has been proposed as one of the main drivers of macroalgal population growth (*Worm & Lotze, 2006*; *Kares et al., 2004*). However, further studies are needed to evaluate spatial and temporal variation of algal propagule supply and subsequent algal settlement in relation to herbivory and nutrient levels. For instance, since herbivores feed on adult macroalgae as well as recruits they might be controlling algal recruitment and abundance at multiple stages of the algal life cycle. Further, different species of herbivores could be important for controlling the same algal species at different life stages as some herbivorous fishes tend to focus more on early successional algae and would be more likely to consumer algal recruits while other herbivorous species focus on late-successional algae (*Burkepile & Hay, 2010*).

The combined effects of herbivore exclosure and nutrient enrichment showed strong effects on abundance of macroalgae on recruitment tiles, particularly during the warmer seasons. This result suggests that reefs that are both overfished (low herbivorous fish biomass) and have high nutrient loading will have higher recruitment of algae during spring and summer. These higher recruitment rates may mean that these reefs are more likely to undergo regime shifts or state changes to communities with abundant algae when corals die. The impacts could be magnified if coral mortality occurs primarily in warmer seasons when herbivorous fishes are the most important for impacting algal recruitment. It is expected that bleaching events caused by increasing water temperature in the Florida Keys will increase in intensity and frequency (*Manzello, 2015*). Severely bleached corals can be quickly colonized by algae (*Diaz-Pulido & McCook, 2002*) that once settled can used existing higher nutrient availability to growth and develop. Thus, protecting herbivores is crucial to facilitate control of macroalgal recruits that can settle on recently dead corals during these summer events. In addition, warmer seasons of the Florida Keys coincide with increases in both DIN and SRP (Appendix S1) which might exacerbate the impact of other ecological stressors during warmest seasons (*Vega-Thurber et al., 2014*). Increases of nitrogen and phosphorus can enhance growth rate of algae and consequently facilitate algal dominance in low herbivory reef. Thus, overfished reef can quickly undergo algal succession with rapid dominance of foliose and filamentous algae, overfished and nutrient enriched reefs can rapidly increase abundance of leathery and articulated calcareous algae when space is available. While filamentous and foliose algae can reduce coral recruitment and harm small adult colonies, leathery and articulated calcareous algae can in addition shade and physically harm colonies by abrasion (*McCook, Jompa & Diaz-Pulido, 2001*). Thus, overfishing herbivores and nutrient pollution can have strong impact on algal succession and, ultimately, their interactions with corals.

## ACKNOWLEDGEMENTS

Many thanks to all those that helped keep the exclosure experiment running: A Shantz, C Pritchard, R Welsh, M Ladd. Thanks to Dr. Deni Rodriguez and Dr. Elizabeth Lacey for their advising during the macroalgae identification process and motivating scientific discussion. We also thank the journal reviewers and editor for their accurate comments that facilitated the quality improvement of the original work.

### Funding

This work was funded by grant OCE 1130786 from the National Science Foundation to DE Burkepile and R. Vega Thurber. The funders had no role in study design, data collection and analysis, decision to publish, or preparation of the manuscript.

### Grant Disclosures

The following grant information was disclosed by the authors:
National Science Foundation: OCE 1130786.

### Competing Interests

The authors declare there are no competing interests.

### Author Contributions

- Alain Duran, Ligia Collado-Vides and Deron E. Burkepile conceived and designed the experiments, performed the experiments, analyzed the data, contributed reagents/materials/analysis tools, wrote the paper, prepared figures and/or tables, reviewed drafts of the paper.

### Field Study Permissions

The following information was supplied relating to field study approvals (i.e., approving body and any reference numbers):
Florida Keys National Marine Sanctuary.
Permits: FKNMS-2009-047 and FKNMS-2011-090.

### Data Availability

The raw data has been supplied as Data S1–S5.

### Supplemental Information

Supplemental information for this article can be found online at http://dx.doi.org/10.7717/peerj.2643#supplemental-information.

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
