# Peer review of "Seasonal regulation of herbivory and nutrient effects on macroalgal recruitment and succession in a Florida coral reef"

_PeerJ, doi:10.7717/peerj.2643_

## Round 0.1 · original submission · Major Revisions

All reviewers feel that the study is very interesting and well written. I recommend that the authors address all the points raised by the reviewer with particular attention to some of the statistical analytical concerns brought by reviewer #2.

·

Basic reporting

Clear English and embedded into relevant literature (some minor corrections necessary). Structure is good and helps the reader to get the message. Figures are well structured and readable (1 minor addition required for Figure 3 to clarify time periods; comments in the PDF).

Experimental design

In general very clear and with a high technical standard. However, for measuring the effects of nutrient enrichment on algae, it is essential to have a regular monitoring program for the nutrient concentrations. From the values the authors report, it does not become clear whether multiple samples were taken and what background variation exists in the area. See comments in the PDF.
Further, I would like to have more details on the settlement tiles (anchorage, angle) and on the quantification technique to determine algae abundance and determination of algal tissue contents. Details on these points as comments in the PDF.

Validity of the findings

Very satisfactory. However, a few more details on the results (such as herbivore data) should be added. Further comments in the PDF.

Additional comments

Dear authors,
it has been very interesting to read your manuscript that tries to disentangle the effects of herbivory and nutrient availability over the season. Please see the comments in the PDF for details on typos, structure and content.
To facilitate overview of the appendices, please structure its content in tables and refer to the corresponding table in the manuscript.

Reviewer 2 ·

Basic reporting

The manuscript is well referenced and well written. I would suggest starting the introduction with some concepts and themes presented in a general enough context that it would be of interest to someone who does not study coral reefs.

Experimental design

Overall this is a well-crafted use of an existing experiment to gain some broad insights. The level of replication is relatively low with n=3 and n=4 for some treatments, but authors still find strong patterns and treatment effects, so this seems sufficient for a least some response variables. Temperature data should be provided for the three seasons.

Validity of the findings

My primary comment is in regards to the analysis and interpretation of the experiment. Since nutrient treatments were applied to the entire plot, and herbivore treatment to quadrats within the plots, I believe this should be analyzed as a split plot ANOVA instead of a basic two factor ANOVA. Also, the various species of algae on a tile are not independent since more of one species probably means there will be less room for another. There is also increased potential for a Type I error with analysis of so many reponses. This would generally be handled with a MANOVA. The main aspect that should be reconsidered is the how to handle the established algal community as a factor. The authors are correct in identifying it as a potential factor in L102-104, but it does not enter into the analysis as an independent factor despite the fact that the results and a previous paper in press show that the established community varies by nutrient and herbivore treatment, and the hypothesis of the present manuscript that propagule supply to tiles is probably affected by the established macroalgal community nearby. So the established macroalgal community varies by plot and is not independent of nutrient and herbivore treatment. There are several ways to handle this. The most straightforward would be to analyze the tile data as if there are four levels of a single factor with each combination of nutrient and herbivore treatments along with its associated macroalgal assemblage as a level. The other way would be to use the established macroalgal community in each quadrat as a covariate for each tile data point. But what variable of established macroalgal community to use as a covariate? Macroalgal richness? Abundance? Identity? Its not clear whether reanalysis would change the basic conclusions of the study, but it would certainly emphasize the potential driving factors and incorporate them in a manner that reflects potential interactions in the field.

Additional comments

This is an interesting study that examines the effects of nutrients, herbivores, and season on macroalgal communities on a coral reef. As pointed out in the introduction, nutrients and herbivores are important drivers of reef dynamics, and many investigators have asked the question of how nutrients and herbivores together can affect the macroalgal dynamics. The authors have emphasized two additional factors that add some ecological relevance to the study of nutrient addition and herbivore exclusions by considering the factors of seasonality and already established algal communities. They have also paired short deployment recruitment tiles with longer deployment succession tiles which is a nice way to understand the development of algal communities.


Other comments:

L4: Why are second and third authors names initialed, but first author not?

L36-37: Effect in spring, but not cooler seasons. What about summer? Presumably summer is warmer than spring, but there is not mention about results in summer in abstract.

L41 is a conclusion with the first mention of abundance of herbivorous fish, and L42 is the first mention of algal growth. Earlier the abstract L32 indicates that algal recruitment, species richness, and species abundance were the responses measured. If fish abundance or algal growth were measured, it should either be established earlier in the abstract or a specific list of responses should not be providedd. Also, were the responses in L32 measured on the tiles, or ongoing experiment? It would be helpful to know at this point if the ongoing experiment is on some kind of tiles as well, or just a section of reef.

The sentence in L102-104 is largely redundant to that in lines L98-102, so one could probably be deleted. The one exception is that the first of those sentences rightfully states that macroalgal abundance is a factor in the experiment, where as the latter sentence and other places in the paper indicate that only herbivory and nutrients are factors being tested. Since a macroalgal community was already established in each of the plots that probably varies by nutrient and herbivore treatment, the dynamics on the tiles is a product of not just the nutrient and herbviory treatments.

L124: Says Zaneveld et al. in press here. Reference cited section says it is in review. Please correct.

L127: Which 3 sides were covered with mesh? 3 sides? Top and two sides? Typically there would be a cage, partial cage, and no cage as three treatments. Could there have been a shading or flow effect in the controls? Is there is a no cage treatment?

L131: Fertilizer tube is reported as 15cm diameter, which seems quite big for that quantity of fertilizer. Especially if they are put side by side, 5 tubes across the plot would be 75 cm of coverage. Maybe it is 1.5cm or 1.5inch diameter tube? But then the 1.5cm diameter hole would also seem big for a smaller size tube.

L138: Please define DIN and SRP at first use.

L175: Was water recirculated within the aquaria, or were they receiving new water. Either way, is it possible that propagules could have entered the aquaria with water?

L181-183: I am not sure I understand the 0.1% designation for individuals. A single individual could grow to cover half the tile. Would that be considered 0.1% or 50%?

L403-417: It should be made more clear which of these results and conclusions are based on results from the present study and which are from previous work. Some of the confusion could be from use of the term overfishing when in fact this study tested the effects of herbivores with cage exclusions which is different than examining the effects of overfishing.

Since temperature is thought to be an important factor that varies seasonally, some temperature data for each of the seasons should be provided.

Reviewer 3 ·

Basic reporting

The paper by Duran et al. describes the results of a factorial experiment conducted to evaluate the influence and interactions of herbivory, nutrients, and existing algal communities on algal recruitment, community structure, and persistence in the Florida Keys.

The paper is well written and provides excellent background. I do suggest the authors take a closer look at prior macroalgal work conducted in Florida to provide better local context for the study. Folks from the Hay lab have worked and published extensively in the keys but the only Hay reference is from another location. Work from Vroom and colleagues from NURC should also be included.

I think the seasonal algal recruitment and relations to established communities is novel and adds to our growing body of knowledge on coral-algal dynamics. That being said, it is a shame that there are data missing from the warmest quarter of the year. While this is stated in the methods (nothing you can do about storms!), I think this should be discussed further and a comment on the potential role of extreme temperatures needs to be discussed (maybe with local or regional references that surveyed in the hot summers)

Finally, I would like to see the authors discuss the correlations between algal recruitment and established communities with caution. The way they describe it now leads the reader to believe the propagules that seed the tiles come from the established plots (local? scale). I suggest they state clearly that the correlations they established do not indicate the source of the propagules and that both the plates and the established communities simply share the same trends. Without knowing what the actual source of the propagules is and what the algal community is beyond the plots sampled prevents the authors from testing this explicitly (correlation is not causation!).

Finally, I do suggest the authors include in their paper photographs of the algal communities on plates and on the reef so that the readers know what these taxa look like.

Again, this is a great study and a well written paper. I look forward to seeing the published product.

Experimental design

very strong design, well explained and analyzed!

Validity of the findings

Very robust and well substantiated conclusions. Please see my general comments on temperature...

Additional comments

Nice job!!!

---

## Round 0.2 · Minor Revisions

The reviewers were pleased with the modifications conducted on the new version. Please, address the remaining minor comments and resubmit a new version.

·

Basic reporting

The revision has been very satisfactory. Good job! Only a few tiny typos should be addressed before acceptance

Experimental design

No further comments. Everything fine

Validity of the findings

No further comments. Everything fine

---

## Round 0.3 · accepted · Accept

The authors have addressed all the reviewers' comments satisfactorily.